# Proteomic Analyses Reveal New Insights on the Antimicrobial Mechanisms of Chitosan Biopolymers and Their Nanosized Particles against *Escherichia coli*

**DOI:** 10.3390/ijms21010225

**Published:** 2019-12-28

**Authors:** Laidson P. Gomes, Sandra I. Anjo, Bruno Manadas, Ana V. Coelho, Vania M. F. Paschoalin

**Affiliations:** 1Chemistry Institute, Federal University of Rio de Janeiro, Avenida Athos da Silveira Ramos 149, Rio de Janeiro 21949-909, RJ, Brazil; 2Instituto de Tecnologia Química e Biológica, Universidade Nova de Lisboa, Av. da Republica, 2780-157 Oeiras, Portugal; varela@itqb.unl.pt; 3CNC-Center for Neuroscience and Cell Biology, University of Coimbra, Rua Larga, 3004-504 Coimbra, Portugal; sandra.isabel.anjo@gmail.com (S.I.A.); bmanadas@gmail.com (B.M.)

**Keywords:** antibiotic effect, anti-microbial molecular mechanism, chitosan polymer, gene ontology, KEGG, metaboanalyst, micro-LC-MS/MS, STRING analysis, ultrasonicated-cs nanoparticles

## Abstract

The well-known antimicrobial effects of chitosan (CS) polymers make them a promising adjuvant in enhancing antibiotic effectiveness against human pathogens. However, molecular CS antimicrobial mechanisms remain unclear, despite the insights presented in the literature. Thus, the aim of the present study was to depict the molecular effects implicated in the interaction of low or medium molecular mass CS polymers and their nanoparticle-counterparts against *Escherichia coli*. The differential *E. coli* proteomes sensitized to either CS polymers or nanoparticles were investigated by nano liquid chromatography–mass spectrometry (micro-LC-MS/MS). A total of 127 proteins differentially expressed in CS-sensitized bacteria were predominantly involved in (i) structural functions associated to the stability of outer membrane, (ii) increment of protein biosynthesis due to high abundance of ribosomal proteins and (iii) activation of biosynthesis of amino acid and purine metabolism pathways. Antibacterial activity of CS polymers/nanoparticles seems to be triggered by the outer bacterial membrane disassembly, leading to increased protein biosynthesis by diverting the metabolic flux to amino acid and purine nucleotides supply. Understanding CS-antibacterial molecular effects can be valuable to optimize the use of CS-based nanomaterials in food decontamination, and may represent a breakthrough on CS nanocapsules-drug delivery devices for novel antibiotics, as the chitosan-disassembly of bacteria cell membranes can potentialize antibiotic effects.

## 1. Introduction

Chitosan (CS) is a well-known polycationic biopolymer that offers multiple benefits in many novel applications, since it allows for complexation with natural antibiotics, antioxidants, proteins and dyes [1,2]. Different forms of chitosan, such as films, capsules and nanoparticles, have been explored by the technoeconomic sector, including drug delivery systems [3], bone tissue engineering [4] and food preservation [1].

Chitosan antimicrobial activity has already been explored on many levels, since it was discovered that its polymers or nanoparticles are effective against Gram positive and Gram negative bacterial strains, including antibiotic-resistant pathogens [5]. In addition, CS has been widely used to carry antibiotic and antimicrobial compounds, such as peptides. In this context, it can serve as a versatile vehicle for the intracellular delivery of therapeutic agents, and is also useful as a non-viral vector for gene delivery [2].

Previous studies have suggested various mechanisms to explain the mode of action leading to chitosan antimicrobial activity [6,7,8].The most explored is the intracellular leakage hypothesis, where bacterial membrane permeability is altered by the interaction between positively-charged-CS and negatively-charged bacterial surfaces, resulting in the loss of cytosolic components through the damaged plasma membrane, leading to cell death [6]. This theory was reinforced by studies investigating CS microparticles, interactions with specific targets hypothesized to be the surface-exposed proteins. Furthermore, it has been demonstrated that CS microparticles likely bind to the outer membrane protein *OmpA* by hydrogen bonding, and to lipopolysaccharides (LPS) via ionic interaction killing the bacteria [9]. In addition, CS antimicrobial activity has been observed, not only in acidic pH, but also at neutral pH (i.e., 7.0) [10].

In a previous study, CS nanoparticles produced by polymer ultrasonication treatment were determined as an antimicrobial compound twice as efficient as chitosan polymer, while also maintaining bactericidal effects at neutral pH. Interactions between CS nanoparticles/polymers and outer membrane were observed on Gram negative bacteria, *Escherichia coli*, indicating a broad antibacterial effect of both CS and its nanosized counterparts [11].

However, the details of the molecular mechanism by which CS nanoparticles/polymers exert their antibacterial effect has not yet been elucidated. A quantitative proteome evaluation followed by functional analyses may provide a comprehensive insight into the global response of bacteria to CS nanoparticles/polymers, which will aid in elucidating the underlying antibacterial mechanism. Clarity around structural proteins, key metabolic pathway enzymes, general stress response proteins and proteins involved in gene expression, such as transcriptional factors at the cellular and subcellular levels, are valuable information concerning the elucidation of CS nanoparticles/polymers effects in the death of bacterial cells.

In this context, the antimicrobial activity of CS nanoparticles/polymers was evaluated through the combination of an integral proteome and bioinformatic analysis using *Escherichia coli* as a model for Gram-negative bacteria. micro liquid chromatography (microLC-MS^3^) combined to the information dependent acquisition (IDA) and sequential windowed data independent acquisition of the total high-resolution mass spectra (SWATH) methods allowed the identification and quantitative analysis of CS-regulated proteins in crude protein extracts prepared from the *E. coli* strain sensitized by CS polymers/nanoparticles. Proteins with differential abundances were identified by multivariate statistical analyses. The differential protein expression signatures in *E. coli* exposed to the CS nanoparticle/polymer were analyzed by STRING in order to propose a putative molecular mechanism for CS action in *E. coli* cells.

## 2. Results and Discussion

### 2.1. Physicochemical Nanoparticle Characterization

Nanoparticles MN and LN were produced by an ultrasonication process from two commercial CS samples, displaying medium (M) and low molecular (L) weights, respectively. Ultrasonic production of chitosan nanoparticles was performed according to our previous study, and a physicochemical characterization was performed [1].

Nanoparticle and polymer size distributions calculated by dynamic light scattering (DLS) and ζ-potential analyses are displayed in Table 1. The 30 min-ultrasonication of M and L chitosan polymers was effective in obtaining nanoparticle counterparts, MN and LN, with reduced hydrodynamic radius, compatible with nanocomposite sizes, but maintaining the electro-kinetic potential, ζ-potential above 30 mV, that represents a much stronger repulsion tendency able to avoid particle agglomeration when compared to suspensions with lower ζ-potential measures [12,13].

Intensity (%) respective to hydrodynamic radius (*Rh*) confirms that ultrasonic (US) irradiation resulted in an increased percentage of smaller molecules in LN and MN samples obtained from commercial CS polymer fragments L and M, respectively.

Mean radius values estimated at 25 °C are also displayed in Table 1. CS molecule radii ranged from 50 to 5150 nm. Before US processing, the L sample presented a major proportion of species with *Rh* of 1219 nm (62%) and a smaller proportion of species presenting *Rh* 4468 nm (38%). US irradiation of L for 30 min generated the LN sample with a proportion of *Rh* values of 468 nm (94%) and 4784 nm (6%), respectively. The M sample is composed of 50% and 18% macromolecules displaying a two-particle population with *Rh* values of 840 and 5130 nm, respectively. After US irradiation for 30 min, this sample yielded smaller MN species, amounting to 84% and 8%, of the two-particle populations with *Rh* of 50 and 355 nm, respectively.

The polydispersity index (PdI) ranges from 0.5% to 1% and indicates the homogeneity of colloidal suspensions, reinforcing the existence of a predominant particle population displaying similar sizes, particularly desirable in the MN and LN samples [13].

### 2.2. Growth Curve Analysis

To characterize the antimicrobial effect, *E. coli* (MAX Efficiency™ DH5α Competent Cells, Invitrogen™) were grown in the presence and absence of the polymers and nanoparticles, M, L, MN and LN, respectively. The cell growth curve indicates a short lag phase followed by exponential growth, representing the log phase, where the middle log phase was achieved at approximately 120 min. Polymers and nanoparticles were added to the middle log phase culture, at three times the minimum inhibitory concentration (MIC) and led a bacteriostatic effect, evidenced by interference in growth continuity or decreased growth rates, as expected for a CS-polymers/nanoparticles antimicrobial effect, displayed in Figure 1.

*E. coli* cells were challenged during the exponential growth phase. Logarithmic growth cultures were chosen for the sensitization test, as they comprise the greatest cell proliferation, metabolic activity, mRNA transcription and protein synthesis period, since cells at high metabolic rates should be the best condition to identify differentially expressed proteins [14].

### 2.3. Differential Proteome Analysis

To identify differentially expressed proteins, an untargeted liquid chromatography–mass spectrometry/mass spectrometry (LC-MS/MS) analysis of total soluble proteins from *E. coli* extracts was performed, obtained from cells lysed after sensibilization by each of the four forms of chitosan, the polymerics L and M and the nanoparticles LN and MN. Protein identification was performed using the Protein Pilot search engine against the *E. coli* O157:H7 Uniprot database. Proteins were quantified based on the sum of transitions from all peptides within the established criteria, in four biological replicates of bacterial cells harvested in the mid-exponential phase for each experimental condition, as displayed in Figure 1. The total number of identified and quantified proteins were 227 and 165, respectively (see Appendix A). Protein identification was made more difficult by the presence of chitosan, due to the interactions established between proteins with each polymer [15]. However, high reproducibility was observed in biological replicates obtained by each exposure condition. Medians of the coefficients of variation for the differentially accumulated proteins in the exposure conditions compared to cells unexposed to CS ranged between 12–21%. The differential proteins were distributed by gene ontology (GO) terms, and protein classes and pathways, and are presented in Appendix A. The most represented metabolic pathways were de novo purine biosynthesis, glycolysis, serine glycine biosynthesis and the tricarboxylic acid (TCA) cycle. That of nucleic acid binding proteins is the most populated class, followed by transferases, oxidoreductases and hydrolases.

Protein abundances were determined by SWATH in CS polymer/nanoparticle-sensitized *E. coli* compared to the control. Proteins in the challenge assays with a fold-change (FC) > 1.67-fold or <0.60-fold and *p* < 0.05 relative to control treatment were considered as differentially expressed. According to the established criteria, a total of 98 unique proteins, 20 less abundant and 78 more abundant than the controls, were identified (Appendix A). Considering each unique protein levels were always altered in the same direction, independently of the polymers/nanoparticles exposure. Among the 78/20 more/less abundant proteins, 46/12, 57/13, 35/8 and 51/10 were identified in samples sensitized with L, LN, M and MN, respectively. A Venn diagram was constructed to indicate the intersection of differentially accumulated proteins in each experimental condition, where Figure 2A indicates the most abundant proteins, and Figure 2B shows the less abundant proteins.

### 2.4. E. coli Proteome Response to Challenge by Chitosan

To depict the mechanism underlining the chitosan polymer/nanoparticles exposure, a focused functional analysis concerning the common seventy-eight most abundant and twenty less abundant proteins under the tested conditions was performed (Appendix A). Sixteen common proteins were accumulated in the four treatments; among them, ten ribosomal proteins (*r-proteins)*: *rplA, rplB, rplI, rplL, rplO, rplS, rplT, rpsA, rpsM* and *rpsQ*. The remaining six non-*r-proteins* also display biological functions directly or indirectly involved in the biosynthesis of proteins and amino acid pathways. *ileS* is an isoleucyl-tRNA synthetase, which occasionally misactivates homocysteine through the catalysis of reactions between isoleucine with organic thiols, forming side chain analogs of homocysteine [16]. *Tsf*, the elongation factor Ts, plays an efficient high policy-folding, which is critical in sequestering interactive surfaces of heterologous proteins from nonspecific protein–protein interactions promoting the formation of inclusion bodies and refolding of unfolded polypeptides [17]. *glyA*, serine hydroxymethyl transferase and *pheA*, bifunctional chorismate mutase/prephenate hydratase, are involved in amino acid biosynthesis, specifically in catalyzing the interconversion between glycine and serine and in the biosynthesis of L-phenylalanine, respectively [18]. *Ppa*, inorganic pyrophosphatase, catalyzes the hydrolysis of inorganic pyrophosphate (PPi) produced during the biosynthesis of the several macromolecules [19]. *ptsI*, phosphoenolpyruvate-protein phosphotransferase, is the major carbohydrate active-transport system and catalyzes the phosphorylation of incoming sugar substrates concomitantly with their translocation across the cell membrane [20]. 

On the other hand, only four less abundant proteins were found after exposure to the four polymers/nanoparticles when compared to control (Figure 2B). Glycerol kinase (glpK) is a key enzyme in the regulation of glycerol uptake and general metabolism, which is phosphorylated to sn–glycerol–3-phosphate that can be catabolized to acetyl-CoA [21]. Stress-induced alternate pyruvate formate–lyase subunit (grcA) is a glycyl radical enzyme that uses pyruvate to produce Acetyl-CoA [22], although its involvement in metabolic pathways is almost unknown. *ompW* is a member of the OmpA–OmpF porin (OOP) family, a receptor for colicin S4, a class of toxic proteins produced by some *E. coli* strains to kill others that lack the corresponding immunity protein in their genome [23]. It is also suggested that OmpW protects bacteria against host phagocytosis [24]. OmpW expression is downregulated upon induction by σ^E^ regulon. σ^E^ activation is triggered by various stress signals, which are sensed in the cellular envelope and communicated to the cytoplasmic compartment by a complex signal transduction pathway [25]. *rbsB* is a periplasmic ribose-binding protein involved in the ATP-dependent ribose uptake [26].

The set of six accumulated proteins (ileS, tsf, glyA, pheA, ppa and ptsI) are included in the same String network when guaA (GMP synthase) is overexpressed in three of the four challenge conditions (L, LN and MN). This is indicative of a direct positive effect on protein biosynthesis or indirect effect through amino acid and nucleotide biosynthesis [27] (Appendix A).

The four proteins displaying decreased levels, after CS-exposure, do not seem to be metabolically related. GlpK, glycerol-phosphate kinase and grcA are involved either directly or indirectly in catabolic processes that originate acetyl-CoA. Decreased expression of these proteins is in line with TCA reduction, as the demand for acetyl-CoA is also reduced [22]. Additionally, nucleotide biosynthesis was expected to be reduced, since ribose uptake is decreased. However, the reported reduction of ompW triggered by stress situations was anticipated [25]. 

### 2.5. E. coli Metabolic and Cellular Processes Globally Affected by Chitosan

To verify if the proteins affected by each CS polymer or nanoparticle were distinct or similar to each other, quantified proteins were submitted to a multivariate statistical analysis according to their contents. The partial least squares-discriminant analysis (PLS-DA) demonstrated that the control condition can be clearly discriminated from all challenge conditions, and that CS nanoparticles were grouped in pairs according with the molecular weight of the original polymer, where, M and MN were clustered in a group, while L and LN were grouped separately, a result reinforced by the significant repeatability within independent samples (*p* < 0.05). The PLS-DA test with five components explained 88% of the variance among the five experimental conditions (see Figure 3).

Thirty-four proteins were found responsible for the discrimination among *E. coli*-exposure conditions with a variable importance in projection (VIP) score above 1 (Appendix A). A total of 85% of them matched those found differentially accumulated by univariate analysis. The validity of the PLS-DA was confirmed by R2 (0.98) and Q2 (estimative of the predictive ability of the model, 0.82) values for the calculated model and by the *p*-value (0.01) determined for the 100 permutation tests.

An analysis of known and predicted functional interactions between the differentially accumulated proteins as determined by the univariate analysis was first performed considering the proteins set identified after exposure to the four chitosan forms, and in the next step, proteins were individually analyzed for each experimental challenge condition compared with the control. STRING (string-db.org), a bioinformatic tool composed of a protein-interaction database, was selected for the analysis using the *Escherichia coli* O157:H7 library as the reference.

To identify the metabolic pathways responsive to CS polymer/nanoparticles sensitization, the accumulated proteins were further analyzed according to the Kyoto Encyclopedia of Genes and Genomes (KEGG) database.

To verify the effect of chitosan on the *E. coli* strain, a global evaluation of the more and less abundant proteins was performed. All quantified proteins were considered as a unique experimental group with no CS polymer/nanoparticle distinction. The global experimental groups were submitted to a STRING protein–protein interaction analysis, according to the group’s contents.

Thus, two overall clustering analyses of proteins identified, such as the most abundant (78) and less abundant (20) were independently performed by STRING. Networks for protein–protein interactions (PPI) are represented in Figure 4. Regarding the most abundant (Figure 4A) and less abundant (Figure 4B) proteins (excluding ribosomal proteins), the relevant protein clusters identified by the analysis are described in Appendix A, eight and one relevant functional cluster were retrieved for the more abundant and less abundant STRING analysis, respectively. These functional clusters include proteins from the purine and pyrimidine and glycolysis/gluconeogenesis metabolisms and the pentose phosphate pathway, in addition to chaperones, and structural and osmoregulatory proteins related to outer membrane integrity. The downregulated protein cluster is formed by *acnB*, *pckA*, *fumA, sdhA* and *sdhB* enrolled into the tricarboxylic acid cycle (TCA). This STRING network also includes the *ompW* gene based on co-expression and experimental/biochemical evidences.

### 2.6. Similarities and Dissimilarities of E. coli Proteome Responses to Chitosan Polymers and Nanoparticles

The STRING functional analysis described in the previous section was reproduced by using a compilation of the proteome results obtained individually for each chitosan sensitization condition. The most affected cell components and metabolic pathways under the four exposure conditions are displayed in Table 2. As expected from the previous global metabolic and cellular processes analyses, the biochemical processes presenting the most significant changes among the four exposure conditions relative to the control were the same previously described in a global comparison between all CS polymers/nanoparticles.

### 2.7. Chitosan, A Destabilizer of E. coli Outer Membrane

To understand the role of chitosan on the outer *E. coli* membrane, it is important to consider that chitosan contains a high content of amino-functional groups, which under favorable conditions, can form electrostatic interactions and adsorb different compounds such as heavy metals, dyes, DNA and proteins [28,29]. The interaction between the protonated chitosan (NH^+3^) and anionic compounds in *E. coli* outer membrane (OM) occurs as a strong electrostatic interaction between the protonated CS and the phospholipid sand lipopolysaccharides (LPS) found exclusively in the asymmetric bilayer of the outer leaflet. The polar nature of LPS is designed to form the outer membrane, a significant barrier to the penetration of lipophilic molecules, and therefore plays an important role in protecting bacteria from various detergents (including bile salts), dyes (including methylene blue) and hydrophobic antibiotics [30].

The LPS inner core region presents a high abundance of phosphate groups, and that makes a significative contribution to the OM negative charge, cooperating in the protein folding processes [31]. Another abundant OM component is the N-terminal domain of OmpA, whose external loops interact with the LPS sugars through hydrogen bonds and salt bridges, and the interdomain linker is flexible, expanding and contracting to pull the globular C-terminal domain up toward the membrane, or push it down toward the periplasm. Structure simulation data suggest a possible OmpA mechanism in maintaining the integrity of the bacterial surface [32], providing cellular mechanical stability [32,33].

Silencing *ompA* expression indicates that the absence of this protein in the outer membrane of the bacteria leads to a reduced antimicrobial CS effect, due to reduced binding of CS-microparticles [9]. Two mechanisms of action were proposed to explain the effect of oleoyl-chitosan nanoparticles on *E. coli*: electrostatic interaction between protonated CS-nanoparticles and the outer membrane components, namely with LPS, which promotes the destabilization of the OM structure, which, after CS nanoparticle internalization, would bind to nucleic acid phosphate groups [34]. Both models explain the drastic reduction in growth rates previously observed in other studies and also reported herein (Figure 1).

A series of reactions must be triggered in order to restructure membrane stability. Computational modeling of global outer membrane protein (OMP) biogenesis suggests the involvement of *skp* [35], a periplasmic chaperone that acts as a regulator for OMP accumulation in the periplasm under stress conditions [36], in particular of OmpA [37]. *Skp* and *ompA* overexpression after exposure to LN/M and L/LN/MN, respectively, seem to be an *E. coli* response in order to maintain OM structure stability (see Table 2). Under these circumstances, *skp* overexpression would enhance the chaperone capacity to improve the insertion of OmpA accumulated in the periplasm into the phospholipid bilayer.

*mdoG* overexpression reinforces that *E. coli* cells can activate an OM stability regulation mechanism, as this periplasmic protein is involved in the synthesis of osmo-regulated periplasmic glucans (OPGs) [38].

In *E. coli*, anionic, highly branched oligosaccharides may accumulate in the periplasmic space in response to stress conditions. The accumulation of CS nanoparticles or the polymers around the *E. coli* OM, visualized by confocal microscopy in a previous study [11], leads to the hypothesis that antimicrobial CS does not only promote a disturbance in OM, but also creates a physical barrier, hindering molecule exchanges between the intra- and extracellular environment. Osmotic imbalance promotes *mdoG* accumulation that results in OPGs overproduction and raises osmotic pressure, which, when persistent for a long period of time, can lead to cell death [38].

The most relevant biochemical processes that occur in *E. coli* exposed to CS-polymers/nanoparticles are depicted in Figure 5. The restructuration of the bacterial membrane and repositioning of CS-bound nucleic acids depends on increased protein biosynthesis along with the formation of all the cell components associated with this process, namely RNA, ribosomes and nucleotides. The proteome analysis reflected the overexpression of proteins and enzymes involved in these biosynthetic pathways, as listed in Table 2. Additionally, a high energetic input coupled to high sugar uptake is needed. As previously reported in similar conditions, a reduction in the complete glycid oxidation through the TCA cycle is replaced by the anaerobic metabolism, known as the Warburg effect, where overall ATP synthesis is reduced due to the incomplete oxidation of substrates [39]. 

The overexpression of *dksA*, a transcriptional factor and regulatory protein, was detected when *E. coli* was exposed to low molecular mass chitosan (L) and nanoparticles (LN and MN). The DskA protein may be related to triggering the *E. coli* modulation of metabolic pathways. The *dskA* participates in the stringent response, a reprogramming of cell metabolism in response to environmental stressors [40,41]. Overall, this stringent response leads to the repression of genes required for rapid growth (such as those involved in rRNA and ribosomal protein biosynthesis), in order to save ATP and precursors for the biosynthetic pathways necessary to maintain life under unsuitable stress growth conditions, such as the activation of genes involved in amino acid biosynthesis, nutrient acquisition and stress survival, in accordance to the results described herein [41].

## 3. Material and Methods

### 3.1. Nanoparticle Preparation

Nanoparticles (MN and LN) were produced by ultrasonication from two commercial chitosan (CS) samples, displaying medium (M) and low molecular (L) weights, with an acetylation degree lower than 25% (Sigma-Aldrich^®^ Co, MO, USA). Samples (2% *w*/*v*) were solubilized in 0.1 M sodium acetate pH 4.0 and irradiated in an ice bath with a BRANSON 450 W model ultrasonic probe (Sonics and Materials Inc., CT, USA) equipped with a 1/2” microtip, at 4 °C for 30 min. The irradiation was performed at 40% amplitude, under a constant duty cycle at 1/1 s intervals.

### 3.2. Nanoparticle and Polymer Characterization

The hydrodynamic radius (Rh), polydispersity index (PdI) and zeta potential (ζ-Potential) measurements pre (M and L) and post (MN and LN) ultrasonic irradiation were determined. The analyses were performed five times for each sample at 25 °C using a ZetaSizer Series Nano-ZS (Malvern Panalytical Ltd., Worcestershire, UK), as described previously [11].

### 3.3. Bacterial Strain and Culture

The *Escherichia coli* DH5α strain was streaked on Luria-Bertani (LB) (Difco™ LB Broth, Miller BD, NJ, USA) agar plates and grown at 37 °C. Cells from a single colony were inoculated into LB broth and cultured overnight in an orbital shaker at 37 °C and 180 rpm. One milliliter of the culture was centrifuged at 5000× *g* for 2 min and the supernatant was discarded. Cellular density was adjusted in a saline solution (0.85% NaCl) (Sigma-Aldrich Co, Spruce St., MO, USA) to turbidity equivalent to McFarland 0.5 standard (1.5 × 10^8^ CFU/mL) [1] and was used at subsequent experiments as the inoculum.

### 3.4. Effects of Antimicrobial Agents on Bacteria

*E. coli* DH5α was inoculated in 10^5^ CFU/mL obtained as described above and the growth curve was constructed by following optical density (OD) measurements at 600 nm, using an Eppendorf BioPhotometer (Eppendorf AG, Hamburg, DEU). Cultures were grown to middle exponential log phase until reaching an optical density of 0.5, when the antibacterial agents M, MN, L and LN were added, in order to reach a final concentration of 3 × the minimum inhibitory concentrations (MIC), corresponding to 1.4, 1.2, 0.9 and 0.9 mg/mL, of CS polymers/nanoparticles respectively [11]. Cell growth was followed for an additional 120 min at 37 °C and 180 rpm. Cells were harvested by centrifugation at 10,000× *g*, 5 min and 4 °C and washed three times with a saline solution (0.85% NaCl). Four biological replicates of each independent growth condition assays were performed, and control samples were assessed without the antimicrobial agent.

### 3.5. Total Protein Extraction

Total proteins extracts were prepared using an extraction protocol for soluble proteins. After the wash step, bacterial pellets were resuspended in 2 mL of 6M urea, 50 mM ammonium bicarbonate (BA) (Sigma-Aldrich Co.) containing an ethylenediaminetetraacetic acid (EDTA)-free protease inhibitor cocktail (cOmplete^TM^, Roche, IN, USA). Cells were disrupted by mechanical lysis using an ultra-sound digital sonicator (W-450, Branson, CT, USA), equipped with a 1/8’’ microtip for 5 min at 10% amplitude, under a duty cycle at 10/10 s (on/off) intervals at 4 °C. Lysates were then spun at 10,000× *g* for 30 min at 4 °C. Total protein concentrations were then estimated in the supernatant using the QuantiPro™ BCA Assay Kit (Sigma-Aldrich Co.), according to manufacturer’s instructions. Blanks were prepared for each independent growth condition assays without the cells. Reproducibility of the protein extract profile among biological replicates was verified by SDS-PAG E on 12.5% acrylamide gels stained with Coomassie R-250 (Merck, Darmstadt, Germany, DEU) [42].

### 3.6. Tryptic Digestion of Total Soluble Protein Extracts

Fifty micrograms of soluble proteins were used for the enzymatic digestion, where the final volume of the reaction was adjusted with a buffer containing 6 M urea and 50 mM ammonia bicarbonate (AB). Green fluorescent protein (MBP-GFP, 2 µg/100 µg total protein) was added to each protein sample as an internal standard for retention time determination and intensity normalization. Reduction of disulfide bounds was performed with ditiotreitol (DTT) (10 mM, 56 °C for 1 h) followed by alkylation of thiol groups with iodoacetamide (30 mM, 25 °C, 30 min in the dark) and at the end a quenching with N-acetil-L-cysteine (37.5 mM, 25 °C, 15 min) was performed. Subsequently, 50 mM AB was added to dilute urea before adding trypsin (1 mg/mL, Trypsin Gold, Mass Spectrometry Grade, Promega, Madison, WI, USA) for protein digestion, followed by incubation at 37 °C for 13 h. The trypsin lysis was interrupted by the addition of 0.5% of formic acid. Tryptic peptides mixtures were dried using a SpeedVac Savant SPD131DDA (Thermo Electron Co., Hopkinton, MA, USA) and resuspended in acetonitrile 2% containing 1% formic acid. Peptide solutions were desalted by C18 microcolumns (OMIX tips, Agilent Technologies, Santa Clara, CA, USA) following the manufacturer’s instruction, vacuum-dried and maintained at −20 °C for liquid chromatography tandem mass spectrometry (LC-MS/MS) assays.

### 3.7. Liquid Chromatography Electrospray Ionization Mass Spectrometry (LC-ESI-MS/MS)

Desalted-peptide pellets were resuspended in 30 µL of 2% acetonitrile (ACN) (Merck, Darmstadt, Germany, DEU) and 0.1% formic acid (FA) (Sigma-Aldrich), and 5 µL of a replicate of each experimental condition were used to create a pooled sample for protein identification. The peptide mixture was then centrifuged at 14,000× *g* for 5 min, to remove the insoluble material, and the supernatant was collected in an adequate vial for LC injection. Four biological replicates per condition, totalizing twenty samples, were submitted to differential proteome analysis.

Samples were analyzed using a Triple TOF^TM^ 6600 System (AB Sciex^®^, Groton, CT, USA) in two steps: information-dependent acquisition (IDA) of the pooled samples and sequential windowed data independent acquisition of the total high-resolution mass spectra (SWATH) acquisition of each individual sample, as described in Appendix A. Peptides were resolved by liquid chromatography (nanoLC Ultra 2D, Eksigent^®^, Kenilworth, NJ, USA) on a MicroLC column ChromXP^TM^ C18CL (300 μm ID × 15 cm length, 3 μm particles, 120 Å pore size, Eksigent^®^) at 5 μL min^−1^ with a multistep gradient: 0–3 min 2% mobile phase B and 3–46 min linear gradient from 2% to 30% B. Mobile phase A consisted in 0.1% FA with 5% dimethyl sulfide (DMSO) (Merck), and mobile phase B, in 0.1% FA and 5% DMSO, in acetonitrile. Peptides were eluted into the mass spectrometer using an electrospray ionization source (DuoSpray^TM^ Source, AB Sciex^®^) with a 25 μm internal diameter (ID) hybrid PEEKsil/stainless steel emitter (AB Sciex^®^).

### 3.8. Proteome Data Processing

A specific library of precursor masses and fragmented ions was created by combining all files from the IDA experiments following the subsequent SWATH assessment. Libraries were obtained using the ProteinPilotTM software (v5.1, ABSciex^®^, Framingham, MA, USA) searching against the UniProt reviewed database of *Escherichia coli* (downloaded at May 2019, 28,900 entries), and MBP-GFP (IS). Data processing was performed using the SWATH^TM^ processing plug-in for PeakViewTM (v2.0.01, AB Sciex^®^, Framingham, MA, USA) [43]. Protein levels were estimated by summing all the transitions from all peptides for a given protein that met the criteria described at Appendix A of an adaptation of [44] and normalized to the total intensity of each sample.

### 3.9. Statistical and Bioinformatics Analyses 

A list of differentially accumulated proteins in each experimental condition was retrieved for the relative quantification for each biological replica (Appendix A). Parameters for the univariate statistical analysis (means, standard deviations, % coefficient of variation (CV), *p*-values for Student’s *t* test and the fold changes were calculated). Proteins identified in assays at abundances of ≥1.9-fold or ≤0.6-fold relative to the control treatment and *p*-value ≤ 0.05 were considered for the network construction using the STRING database for the *Escherichia coli* O157 strain (4522 protein entries) (STRINGdb, [45] with an average confidence of 0.4. KEGG pathways, Uniprot and NCBI enrichment analysis were accessed thought STRINGdb. To evaluate any clustering behavior through a multivariate analysis, an unsupervised Principal Components Analysis (PCA) was applied to all data sets using the Metaboanalyst Analytical Pipeline [46]. The Gene List Analysis tool from the Panther Classification System [47] provided protein classification by gene ontology terms, protein class and pathway. 

## 4. Conclusions

In summary, proteome methods applied to detect the overexpressed and underexpressed proteins by *E. coli* exposed to CS nanoparticles successfully point to the cell membrane as the main CS nanoparticle/polymer target. Induction of the biosynthetic pathways of amino acids and purine nucleotides were observed, repeating a pattern previously reported when bacteria cells are under stress conditions. The interaction between the CS polymer/nanoparticle to the outer membrane components, namely LPS, activates the cellular response promoted by DksA. DksA participates in the stringent response, a reprogramming of cell metabolism in response to CS sensitization. The repression of genes required for rapid growth redirect the protein biosynthesis to a few, however, indispensable, proteins. The reduction in the overall ATP synthesis due to the incomplete glucose oxidation occurs because the TCA cycle is replaced by the anaerobic metabolism, in order to conserve energy to be used in the formation of precursors for the biosynthetic pathways necessary to maintain life under unsuitable stress growth conditions. Nucleotide and amino acid biosynthesis are diverted to promote the translation of a select number of proteins, among them repair enzymes and structural proteins.

Chaperone-like proteins, such as the periplasmic Skp, are formed to assist unfolded proteins in repairing stress-induced molecular damages in *E. coli*, particularly the insertion of OmpA in the outer cell membrane against the disassembly caused by the interaction of CS-polymers/nanoparticles to LPS. If not successful, a structural disruption of bacterial membrane might occur and eventual cell death.

Furthermore, a proteome differential analysis can be applied to explore the molecular delivery mechanisms of bioactive agents encapsulated in chitosan nanoparticles to target cells. The transitory disassembly of the outer membrane, if well controlled, may aid in the delivery of the bioactive agents to target cells. This study provides valuable knowledge for further investigations on the bactericidal activity of CS nanoparticles/polymers in Gram positive bacteria and open the road for the design of CS-nanodevices displaying broad bactericidal activity.

## Figures and Tables

**Figure 1 ijms-21-00225-f001:**
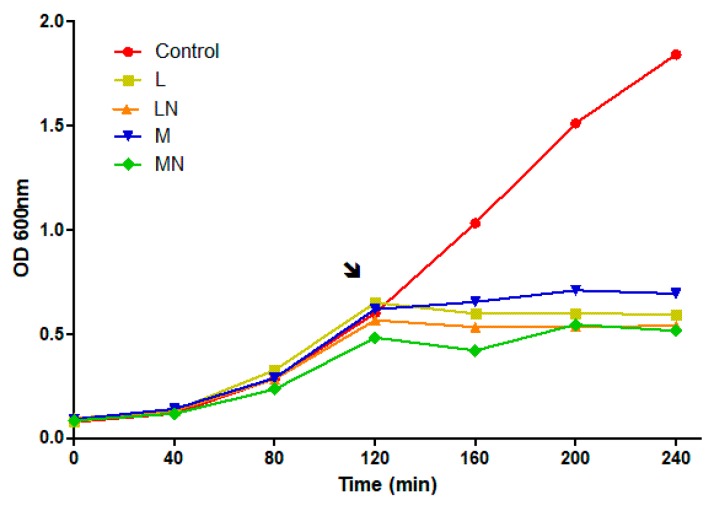
*Escherichia coli* growth curve in the presence of chitosan polymers and nanoparticles. The DH5α strain was grown in Luria–Bertani medium at 37 °C and 180 rpm. The black arrow indicates the addition of medium (M), and low molecular (L) polymers or MN LN nanoparticles in order to reach final concentration of 3 × minimum inhibitory concentration (MIC) values, of 1.41, 0.9, 1.2 and 0.9 mg/mL, of CS polymers/nanoparticles respectively.

**Figure 2 ijms-21-00225-f002:**
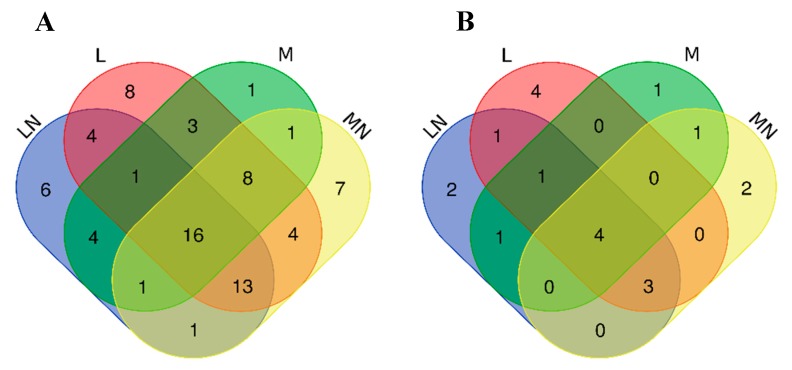
*E. coli* challenge-responsive proteins under L (red), LN (blue), M (green) and MN (yellow) exposure represented by Venn diagrams: (**A**) number of differentially expressed, more abundant proteins and (**B**) number of differentially expressed, less abundant proteins among the four treatments, as determined by liquid chromatography–mass spectrometry (LC-MS/MS). Complete list of proteins is presented in Appendix A, and the lists of each intersection are indicated in Appendix A.

**Figure 3 ijms-21-00225-f003:**
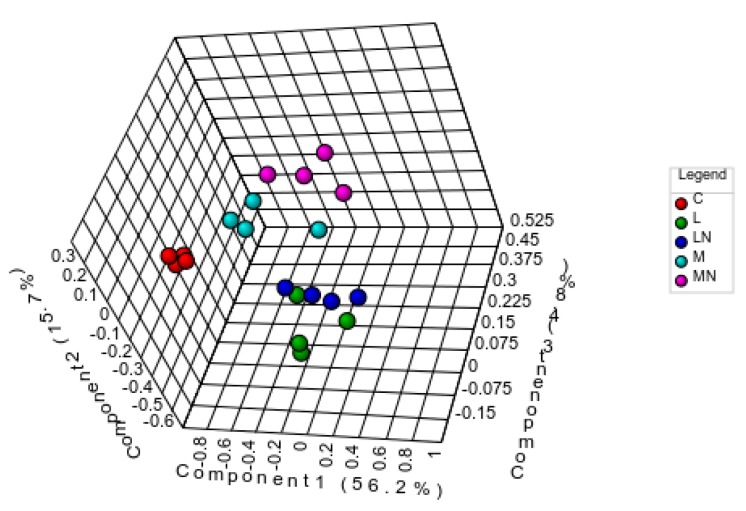
Partial least squares-discriminant analysis (PLS-DA) test (PC = 5) of all nano-LC-MS/MS quantified proteins. The 3D-plot shows biological replicates distribution considering the first three principal components. Control (C), L, LN, M and MN conditions are represented in red, green, dark blue, light blue and pink, respectively. Areas correspond to a 95% confidence region. Dots within each area represent the biological replicates of each exposure condition (*n* = 4).

**Figure 4 ijms-21-00225-f004:**
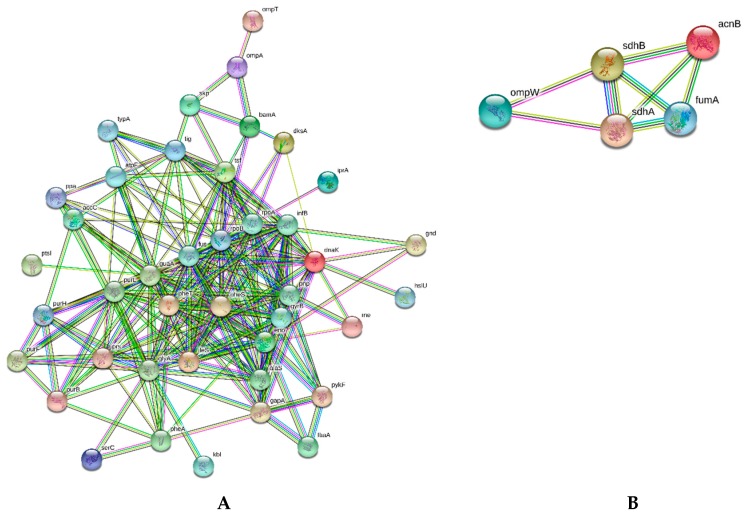
Graphical representation of the STRING analysis without disconnected nodes in the network. (**A**) represents the overall more abundant proteins excluding ribosomal proteins (number of edges is 1152; protein–protein interaction (PPI) enrichment *p*-value < 1.0 × 10^−16^), (**B**) represents the network of less abundant proteins (number of edges is eight; PPI enrichment *p*-value < 9.0 × 10^−6^). Protein–protein interaction identified under all experimental condition are, listed in Table 2, using 0.4 as the minimum score required for interaction, taking into account parameters concerning active interaction sources, such as bibliographic citations, data co-expression, chromosomal neighborhood, genetic fusion and co-occurrence.

**Figure 5 ijms-21-00225-f005:**
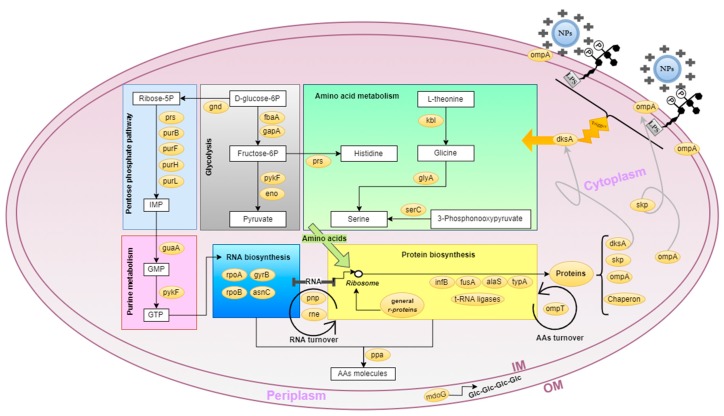
Schematic diagram of upregulated *E. coli* proteins and pathways in response to CS nanoparticles/polymers sensibilization. Arrows indicate positive regulation, yellow circles represent upregulated proteins; NPs, nanoparticles; LPS, Lipopolysaccharides; IM, inner plasmatic membrane; OM, outer plasmatic membrane; Glc, glucans and AAs amino acids. The proteins genes comprise the chaperone group, *bamA, dnaK, hslU*, *tig* and *skp*; t-RNA ligases; *ileS, pheS, pheT* and tsf and general r-proteins, *rplA, rplB, rplC, rplE, rplF, rplI, rplJ, rplK, rplL, rplM, rplO, rplS, rplT, rplU, rplV, rpmA, rpmB*, and *rpmD.*

**Table 1 ijms-21-00225-t001:** Physicochemical characterization of chitosan (CS) samples estimated before (polymers, shaded columns) and after ultrasound irradiation (nanoparticles, non-shaded columns).

Sample	L		LN		M		MN	
ζ potential (mV)	50.2 ± 1.9		40.1 ± 3.6		40.1 ± 1.1		33.1 ± 2.8	
*Rh* (nm)	1219	4468	468	4784	840	5132	355	50
Intensity (%)	62	38	94	6	50	18	84	8
PdI	0.55 ± 0.02		0.47 ± 0.01		0.99 ± 0.01		0.78 ± 0.02	

Zeta potential (ζ potential), hydrodynamic radius (*Rh*), intensity percentage of light scattering (Intensity), polydispersity index (PdI) of chitosan polymers L and M—polymeric commercial CS with low and medium molecular masses. LN and MN—nanoparticles generated from low (L) and medium molecular (M) weights fragmented by 30 min-ultrasonication.

**Table 2 ijms-21-00225-t002:** *E. coli* genes/proteins, metabolic pathways and cell components affected by chitosan exposure and identified by STRING. The genes encoding differentially expressed proteins were listed for each CS-polymer/nanoparticle, see Appendix A for detailed information.

	L	LN	M	MN
**Overexpressed**				
**STRING Parameters**				
Edges	420	675	280	598
PPI	<1.0 × 10^−16^	<1.0 × 10^−16^	<1.0 × 10^−16^	<1.0 × 10^−16^
Purine metabolism	*guaA, pnp, purB, purL, pykF, gnd, rpoA*	*guaA, pnp, prs, purB, purH, gnd, rpoA, rpoB*	*guaA, prs, pykF*	*prs, purB, purF, purH, gnd, rpoA*
Amino acid biosynthesis	*fbaA, glyA, pheA, pykF, kbl*	*eno, gapA, glyA, pheA, prs, serC*	*glyA, pheA, prs, pykF*	*eno, fbaA, glyA, pheA, prs*
Outer membrane proteins	*accC, dnaK, dksA, hslU, ompA, ompT, tig*	*accC, dksA, mdoG, ompA, ompT, skp*	*accC, hslU, infB, mdoG, skp, tig*	*accC, bamA, dksA, infB, ompA, ompT*
**Underexpressed**				
**String Parameters**				
Edges	3	8	0	1
PPI	0.0111	6.92 × 10^−16^	−	0.366
TCA	*sdhA, sdhB*	*acnB, fumA, sdhA, sdhB*	−	*pckA, sdhB*

Protein identification was performed using the Protein Pilot search engine against the *E. coli* Uniprot database. Proteins were quantified based on the sum of transitions from all peptides within the established criteria, in the four independent biological replicates of soluble protein extracts from bacterial cells lyses after the sensitization test for each CS-polymer/nanoparticles sensitization, namely LN and MN—nanoparticles generated from low molecular (L) and medium molecular (M) weights fragmented by 30 min-ultrasonication. No accumulation differences are represented by (–), PPI—protein–protein interactions and TCA-tricarboxylic acid cycle.

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
