# Peer review of "Proteomic Analyses Reveal New Insights on the Antimicrobial Mechanisms of Chitosan Biopolymers and Their Nanosized Particles against Escherichia coli"

_ijms, 2019, doi:10.3390/ijms21010225_

Round 1

Reviewer 1 Report

Evaluation of “Proteomic analyses reveal new insights on the antimicrobial mechanisms of chitosan biopolymers and their nanosized particles against Escherichia coli” by Gomes et al.

The paper deals with the study of the molecular mechanisms implicated in the interaction of low or medium molecular mass CS polymers and their nanoparticle-counterparts against Escherichia coli. The topic is very interesting and presents very well supported results. However, there are several aspects that need to be addressed before it can be ready for publication.

Abstract

Lines 17-18: I suggest including that there are some papers that present some insight, but it is still very confusing or unclear.

Keyword: Alphabetical order

Introduction

Page 2 lines 1-5: There are several theories, but I do not think authors should highlight that there is one which is the most accepted since there are several events that could occur at the same time. Please check the next paper and include it in your discussion: Chitosan coatings enriched with essential oils: Effects on fungi involved in fruit decay and mechanisms of action. Trends in Food Science & Technology, 78, 61-71.

Results and discussion

Line 42: remove the dot after the word sample.

It is important clarifying from the beginning which are the samples treated with ultrasound and for what reason. If this aspect is left for the material and methods section it will be confusing.

Author Response

Answers to Reviewer 1

Abstract

Answer: In line 2, the sentence: “However, the CS antimicrobial mechanisms remain unclear.” was changed to: “However, the molecular CS antimicrobial mechanisms remain unclear, despite the insights presented in the literature”

Lines 17-18: I suggest including that there are some papers that present some insight, but it is still very confusing or unclear.

Answer: The abstract was modified, as suggested.

Keyword: Alphabetical order

Answer: The keywords were reorganized in alphabetical order, as suggested, and now read: Gene Ontology; Hydrodynamic radius; KEGG; Light scattering; Metaboanalyst; Nano-LC-MS/MS; PLS-DA test; STRING analysis; Ultrasonicated-CS nanoparticles; Venn diagram

Introduction

Page 2 lines 1-5: There are several theories, but I do not think authors should highlight that there is one which is the most accepted since there are several events that could occur at the same time. Please check the next paper and include it in your discussion: Chitosan coatings enriched with essential oils: Effects on fungi involved in fruit decay and mechanisms of action. Trends in Food Science & Technology, 78, 61-71.

Answer: This reference was included, as suggested: reference 2, page 1, line 39.

Results and discussion

Answer: Line 42: remove the dot after the word sample.

It is important clarifying from the beginning which are the samples treated with ultrasound and for what reason. If this aspect is left for the material and methods section it will be confusing.

Answer: The dot after the word sample was removed.

Answer: The reviewer is correct. A sentence was included at the beginning of the Results and Discussion section 2.1, Physicochemical Nanoparticle Characterization page 2 lines 37-40.

“Nanoparticles MN and LN were produced by ultrasonication process from two commercial CS samples, displaying medium (M) and low molecular (L) weights, respectively. Ultrasonic production of chitosan nanoparticles was performed according to our previous study and a physicochemical characterization was performed”.

Reviewer 2 Report

There is a lot of information in the manuscript, however, there are some suggestions for authors. Thank you very much!

In part of the introduction, the author maybe can write more information for Chitosan and Escherichia coli.  In part of the conclusions, there is not really clear meaning for the research, the author can write more and obvious sentences to present it for the manuscript. In the manuscript, all of the data need to reset it up. There are quite mess up to read the manuscript.

Author Response

Answers to reviewer 2

General Comments by the authors:

We believe that we have fully addressed all the reviewer’s concerns and comments.

The entire manuscript was revised and all modifications were highlighted in red.

The suggested modifications have polished the manuscript and increased its overall impact. We would like to thank the reviewer for his/her insight and thoughtful critiques.

After the insertion of suggested modifications by the reviewers, the entire text was revised by an editing specialized company in order to improve English grammar and syntax.

Reviewer comments precede our responses.

In part of the introduction, the author maybe can write more information for Chitosan and Escherichia coli. In part of the conclusions, there is not really clear meaning for the research, the author can write more and obvious sentences to present it for the manuscript. In the manuscript, all of the data need to reset it up. There are quite mess up to read the manuscript.

Answer: The entire manuscript was revised in order to clarify the presented results.

Obvious sentences were introduced at the beginning of each section, in order to direct the reader to the results presented in each section, as suggested.

Conclusions were improved, adding the new insights we obtained about protein availabilities that could aid in better underlining the molecular mechanisms involved in antimicrobial chitosan effects.